# 7-Chloroquinolinehydrazones as First-in-Class Anticancer Experimental Drugs in the NCI-60 Screen among Different Investigated Series of Aryl, Quinoline, Pyridine, Benzothiazole and Imidazolehydrazones

**DOI:** 10.3390/ph16050691

**Published:** 2023-05-03

**Authors:** Georgiana Negru (Apostol), Alina Ghinet, Elena Bîcu

**Affiliations:** 1Faculty of Chemistry, ‘Alexandru Ioan Cuza’ University of Iasi, Bd. Carol I, nr. 11, 700506 Iasi, Romania; georgiana_16_95@yahoo.com; 2Junia, Health and Environment, Laboratory of Sustainable Chemistry and Health, F-59000 Lille, France; 3Institut National de la Santé et de la Recherche Médicale, Centre Hospitalier Universitaire de Lille, Institut Pasteur Lille, U1167—RID-AGE—Facteurs de Risque et Déterminants Moléculaires des Maladies Liées au Vieillissement, University of Lille, F-59000 Lille, France

**Keywords:** hydrazone, 7-chloroquinoline, NCI panel, antitumor, pharmacomodulation, SAR

## Abstract

In the context of a continuously increasing global cancer risk, the search for new effective and affordable anticancer drugs remains a constant demand. This study describes chemical experimental drugs able to destroy cancer cells by arresting their growth. New hydrazones with quinoline, pyridine, benzothiazole and imidazole moieties have been synthesized and evaluated for their cytotoxic potential against 60 cancer cell lines. 7-Chloroquinolinehydrazones were the most active in the current study and exhibited good cytotoxic activity with submicromolar GI_50_ values on a large panel of cell lines from nine tumor types (leukemia, non-small cell lung cancer, colon cancer, CNS cancer, melanoma, ovarian cancer, renal cancer, prostate cancer and breast cancer). This study provided consistent structure-activity relationships in this series of experimental antitumor compounds.

## 1. Introduction

The burden of cancer continues to grow globally. In addition to quality and timely diagnosis, access to adapted treatments remains an important priority in order to increase the survival rate of many types of cancer and defeat cancer [1]. The discovery of new anticancer experimental drugs remains a major issue in medical chemistry. 

7-Chloroquinoline hydrazones have been previously developed and successfully tested in vitro for many biological activities [2,3,4,5,6,7,8,9,10,11,12,13,14,15,16,17,18,19,20,21,22,23,24,25,26,27,28,29,30,31,32,33]. The major part of the studies has been carried out to identify their antimalarial properties [2,3,4,5,6,7,8,9,10]. In addition, these hydrazones were explored for their anti-leishmanial [10,11,12,13], anti-tubercular [3,14,15,16,17,18], anti-bacterial [19,20], anti-oxidant [21,22], anti-fungal [23,24], anti-viral [25], anti-inflammatory [26], anti-prion disease [27], anti-convulsant [28], antinociceptive [29], but also anticancer activities [30,31,32,33]. Many chemical modulations have been realized on 7-chloroquinoline hydrazones. Most of the reported derivatives contain a differently substituted aryl unit, and fewer derivatives have another heterocycle in their structure. In search of new anticancer agents, in this study, we were interested in obtaining 7-chloroquinoline hydrazones with anti-proliferative activity against several cancer cell lines and establish a structure–activity relationship (SAR) in these series (Figure 1). The most promising 7-chloroquinoline hydrazones as anticancer agents reported so far are presented in Figure 1. Hydrazone I inhibited the growth of SF-295 CNS cancer cells with an IC50 value of 0.688 µg/cm^−3^ (Figure 1) [30]. Only one report described molecules containing a heterocyclic unit in addition to the 7-chloroquinoline ring as promising anti-proliferative agents, with notable cell growth inhibition of melanoma MDA-MB-435 cells [33]. The best antitumor activity was obtained using a pyrrole unit (compound **II**, Figure 1). The latter compound was four times more active than reference Doxorubicin [33]. This work aims to explore additional heterocyclic units for the improvement of the antitumoral activity of final compounds on the NCI-60 cancer cell line panel. Three types of chemical modulations have been envisaged on the target compounds **1**–**27**, first on the 7-chloroquinoline unit (A-ring): (1) the importance of the position of the chloro substituent and (2) the replacement of the chloroquinoline unit by another heterocycle (pyridine, benzothiazole and dihydroimidazole) or by a substituted phenyle (Figure 1); and finally, (3) aromatic and heteroaromatic (phenothiazine, indole and thiophene) units have been designed for the B-ring of target molecules **1**–**27** (Figure 1 and Figure 1). 

## 2. Results and Discussion

### 2.1. Synthetic Strategy

A series of aldehydes **28**–**39** was reacted with substituted (hetero)arylhydrazines **40**–**49** in ethanol at reflux. All starting aldehydes were commercially available except aldehyde **34**, which was obtained in 70% yield by formylation of *N*-methylphenothiazine **50** with Vilsmeier reagent in dichloroethane [34,35]. Final condensation products were easily obtained in good yields (54–84%) and confirmed as hydrazone derivatives **1**–**27** as *E*-isomers (Figure 1). 

### 2.2. Biological Evaluation

All synthesized compounds **1**–**27** have been submitted to the National Cancer Institute (NCI) (Germantown, MD, USA) and have been selected for evaluation in the NCI-60 cell-line screen. Compounds are generally selected for screening based on their ability to add diversity to the NCI small molecule compound collection [36]. The NCI-60 panel includes cell lines from nine tumor types (leukemia, non-small cell lung cancer, colon cancer, CNS cancer, melanoma, ovarian cancer, renal cancer, prostate cancer and breast cancer) and is extremely well characterized at the molecular level with both in-house and crowd-sourced characterization, including exome sequence for mutations, SNPs, DNA methylation, metabolome, mRNA, microRNA, and protein expression. This molecular characterization dataset enables interrogation of patterns of growth inhibition by the investigational drug set looking for characteristics of the cell lines that determine sensitivity. 

Selected compounds **1**–**27** have been tested initially at a single high dose (10 µM) in the full NCI 60-cell panel. The one-dose data is reported as an average of the percent growth of treated cells. The number reported for the single dose (10 µM) test is the cell growth relative to the no-drug control and relative to the cell count at time zero. This allows the detection of both cell growth inhibition (values between 0 and 100, cytostatic effect) and cell lethality (values less than 0, cytotoxic effect). The same screening methodology is applied for the 5-dose test, also realized by NCI on the most promising compounds identified in the one-dose test. To the best of our knowledge, this is the first study of 7-chloroquinolinehydrazones on such a large number of cancer cell lines. Among tested compounds **1**–**27** in the one-dose screen, seventeen hydrazones **1**, **2**, **6**, **7**, **9**, **12**–**20**, **22**, **23** and **25** were the most active, while the others did not have any impact on cancer cell growth. The cell growth inhibition values induced by these compounds is presented in Table 1. Interestingly, six compounds, **6**, **13**, **16**, **20**, **23** and **25**, displayed significant cytotoxic effects in the one-dose screen on almost all tested cell lines (Figure 2). However, not only these molecules have progressed to the full 5-dose assay. A total of ten compounds, **6**, **7**, **13**–**16**, **18**, **20**, **23** and **25**, with both promising cytostatic and cytotoxic effects, have been further selected by the NCI for 5-dose in vitro human cancer cell growth inhibition in order to obtain their GI_50_ values (Table 2). Compounds **6**, **7**, **14**, **15** and **18** displayed GI_50_ values in the micromolar range, while compounds **13**, **16**, **20**, **23** and **25** were more active with GI_50_ values in the submicromolar range up to 120 nM (*e.g*., hydrazone **16** effect on SR leukemic cells, Table 2). The most effective cancer cell growth inhibition has been achieved with hydrazone **23** (Table 2). Therefore, several structure-activity relationships could be determined. 

### 2.3. Structure-Activity Relationships 

Dihydroimidazole derivatives **3**–**5** were completely ineffective and did not deserve future development as anticancer agents. In the same way, hydrazones bearing two differently substituted aryl units **1** and **2** failed to achieve effective cancer cell growth inhibition (Table 1). The combination of two heterocyclic units was highly more tolerated (e.g., 7-chloroquinoline and 1-methyl-5-methoxyindole in the structure of hydrazone **23**, Figure 1 and Table 2) and proved to be the best pharmacomodulation in the current study. Additional chemical modulations were performed on these specific heterocyclic units.

Concerning the *1*-methyl-5-methoxyindole unit (B-ring of target compounds): (1) the replacement of the indole unit with an aromatic unit (3-chloro-4-methoxyphenyl in hydrazone **6** and 4-hydroxy-3,5-dimethoxyphenyl in hydrazone **7**) conserved antitumor efficiency (GI_50_ values in the micromolar range) but resulted in diminished antitumor effect compared to indole-hydrazones **16** and **23** (Table 2). Hydrazones **6** and **7** displayed similar overall efficiency, and only two differences have been observed on SR (leukemia) and SK-MEL-2 (melanoma) cell lines, hydrazone **6** being slightly more active (**6**: GI_50_ (SR) = 0.7 µM vs. **7**: GI_50_ (SR) = 1.9 µM and **6**: GI_50_ (SK-MEL-2) = 1.79 µM vs. **7**: GI_50_ (SK-MEL-2) = 16.3 µM, Table 2); (2) the replacement of the indole unit by a 10-methyl-phenothiazine group in hydrazone **8** was unfavorable to the biological activity, and no inhibition effect was obtained with the latter compound. In order to rule out the potential influence of the overall volume of molecule **8** on the lack of biological effect, smaller-size phenothiazine hydrazones have been synthesized (hydrazones **9**–**12**, Figure 1). None of these hydrazones were active on the NCI-60 screen. Some modest cytostatic effects were however registered in the one-dose screen with hydrazone **9**, decorated with a 4-methoxyphenyl unit, and hydrazone **12**, decorated with a 3-bromophenyl unit (Table 1); (3) the suppression of the methyl substituent on the indole nitrogen was tolerated but resulted in slightly reduced overall cancer cell growth inhibition (compare GI_50_ values of methylated-compound **23** vs. non-methylated analog **16**, Table 2); finally, (4) the importance of the 5-methoxy substituent on the indole ring was also explored. The absence of the 5-methoxy substituent in hydrazone **13** conserved the antitumor efficiency on a large number of cell lines but displayed slightly reduced efficiency compared with 5-methoxy-analogue hydrazone **16**, especially on ovarian, renal, prostate and breast cancer cell lines (Table 2). A similar activity profile has been observed with hydrazone **20** compared to 5-methoxy analog hydrazone **23**, underlining the added value of the 5-methoxy group (Table 2). Finally, the replacement of the electro-donating M+ methoxy group (hydrazone **16**) by an electro-withdrawing -I bromo substituent in hydrazone **25** conserved the antitumor effect on a large number of cancer cell lines and showed some decreased effect on ovarian, renal, prostate and breast cancer cell lines (compare hydrazones **16** and **25**, Table 2).

The chemical modulations performed also concerned the 7-chloroquinoline unit (A-ring of target compounds): the replacement of the 7-chloroquinoline unit in hydrazones **13**, **16**, **20** and **25** by a chloro- or bromopyridine moiety in hydrazones **14**, **15**, **18**, **19**, **21**, **22**, **26** and **27** resulted in global diminished antitumor activity but displayed some interesting effects on CNS (**14**: GI_50_ (SNB-75) = 0.14 µM; **15**: GI_50_ (SNB-75) = 0.38 µM) and breast (**14**: GI_50_ (BT-549) = 0.24 µM; **15**: GI_50_ (BT-549) = 0.17 µM) cancer cells and were slightly more active than 7-chloroquinoline congener **13** (GI_50_ (SNB-75) = 1.38 µM; GI_50_ (BT-549) = 1.55 µM) (Table 2). The same particularity was observed between hydrazones **16** (with a 7-chloroquinoline unit) and **18** (with a 5-bromopyridin-2-yl group). Despite slightly diminished cell growth inhibition potential on the major part of tested cell lines, hydrazone **18** inhibited more effectively the growth of SNB-75 and BT-549 cells compared to hydrazone **16** (**16**: GI_50_ (SNB-75) = 1.58 µM; **18**: GI_50_ (SNB-75) = 0.17 µM); **16**: GI_50_ (BT-549) = 1.76 µM; **18**: GI_50_ (BT-549) = 0.17 µM) (Table 2). Compound **19** did not enter 5-dose testing and was not able to be compared with related analogs **16** and **18**. Based solely on the data from the one-dose screen (Table 1), compound **19** with a 6-chloropyridin-2-yl group seemed slightly less active than compound **18** with a 5-bromopyridin-2-yl group. Going further with the comparison of 7-chloroquinoline hydrazone **20**, bromopyridine hydrazone **21** and chloropyridine hydrazone **22**, in this case, the activity of the first compound **20** was largely superior to that of analogs **21** and **22**, underlining the importance of the 7-chloroquinoline unit when the other part of the molecule was an *N*-methylindole. Compounds **21** and **22** did not satisfy pre-determined threshold inhibition criteria in a minimum number of cell lines and consequently did not progress to the full 5-dose assay. This was also supported by comparing the activity of hydrazone **25** with that of hydrazones **26** and **27**. While the 7-chloroquinoline hydrazone **25** showed one of the best cancer cell growth inhibitions in the current study, bromopyridine hydrazone **26** and chloropyridine **27** were completely inactive. Two additional heterocyclic units have been used to substitute the 7-chloroquinoline unit. First, the replacement by a benzothiazole ring in hydrazone **17** was not tolerated and abolished the antitumor effect compared to 7-chloroquinoline hydrazone **16** (Table 1 and Table 2). Finally, the importance of the 7-chloroquinoline moiety was once again demonstrated by using a non-chlorinated quinoline in hydrazone **24** (Figure 1).

The absence of the chlorine atom concomitant with the change of the substitution position on the quinolinic ring (from 4 in hydrazone **23** to 2 in hydrazone **24**) was detrimental to the antitumor activity; this compound being stopped in the one-dose screen for lack of activity. 

A global visualization of the structure-activity relationships for hydrazones obtained in this study has been proposed in Figure 3.

### 2.4. Conclusions

To conclude, twenty-seven hydrazones **1**–**27** were synthesized in this work and screened for their potential to inhibit the NCI-60 cancer cell panel. As oncology treatment moves toward personalized, targeted therapeutic agents, the NCI-60 human tumor cell line panel remains an ideal community-wide tool to further understand the disease targets of new agents. Two main points of chemical modulation of the model molecules **I** and **II** (Figure 1) have been studied and were realized on their A and B rings. This allowed to establish several structure-activity relationships in this series. The best pharmacomodulation was obtained by associating two heterocyclic units on the hydrazone bridge: 7-chloroquinoline as the A-ring and 1-methyl-5-methoxyindole as the B-ring. The resulting hydrazone **23** showed very promising antitumor activity on the NCI-60 cancer cell lines (GI_50_ values in the submicromolar range). While the 1-methyl-5-methoxyindole tolerated different substitutions (suppression of the methyl, substitution or suppression of the 5-methoxy group), the 7-chloroquinoline ring was found to be essential for antitumor activity. Interestingly, promising cell growth inhibition was detected for chloropyridine and bromopyridine hydrazones **14**, **15** and **18** on SNB-75 (CNS cancer) and BT-549 (breast cancer) cell lines. These compounds outperformed the most active hydrazone **23** in this study on these particular cancer cell lines and deserved future biological investigation. 

The study resulted in a new collection of 7-chloroquinoline hydrazones with promising potential for further development of compounds for oncology. Their mechanism of action is currently under study and will be reported in due course.

## 3. Materials and Methods for Synthesis and Characterizations

Starting materials are commercially available and were used without further purification (suppliers: Carlo Erba Reagents S.A.S., Val de Reuil, France, Thermo Fisher Scientific Inc., Illkirch-Graffenstaden, France, and Sigma-Aldrich Co., Saint-Quentin-Fallavier, France). Nuclear magnetic resonance (NMR) spectra were acquired at 500 MHz for ^1^H-NMR and at 125 MHz for ^13^C-NMR on a Bruker Avance III 500 MHz spectrometer (Bruker, Mannheim, Germany) with tetramethylsilane (TMS) as internal standard, at room temperature (RT). All spectra have been realized using deuterated solvents (CDCl_3_ 99.8%D + 0.03% TMS *v*/*v* or DMSO-d_6_ 99.8%D + 0.03% TMS *v*/*v*), purchased from Eurisotop, Saint-Aubin, France. The calibration has been realized using TMS pic as the 0.00 ppm value in the registered spectra. Chemical shifts (δ) are expressed in ppm relative to TMS. Splitting patterns are designed: s, singlet; d, doublet; dd, doublet of doublets; t, triplet; q, quadruplet; quint, quintuplet; m, multiplet; sym m, symmetric multiplet; br s, broaden singlet; br t, broaden triplet. Coupling constants (*J*) are reported in Hertz (Hz). ^1^H and ^13^C NMR spectra for all synthesized hydrazones **1**–**27** are provided in the Appendix A section. Melting points were measured on an MPA 100 OptiMelt^®^ apparatus (Stanford Research Systems, Sunnyvale, CA, USA) and are uncorrected. Thin layer chromatography (TLC) was realized on Macherey Nagel silica gel plates (Macherey Nagel, Hœrdt, France) with fluorescent indicator and were visualized under a UV lamp at 254 nm and 365 nm. IR spectra were recorded on an FTIR Bruker Tensor 27 Spectrometer. Elemental analyses (C, H, N) of new compounds were determined on a Thermo Electron apparatus (Thermo Fisher Scientific Inc., Illkirch, France) by “Pôle Chimie Moléculaire-Welience”, Faculté des Sciences Mirande, Dijon, France.

### 3.1. Synthesis of 10-Methyl-10H-phenothiazine-3-carbaldehyde (***34***)

The Vilsmeier reagent was generated by stirring a mixture of DMF (11.64 mL, 8 equiv.) and POCl_3_ (5.8 mL, 4 equiv.) at 0 °C for 30 min. Then, a solution of 10-methyl-10*H*-phenothiazine (4 g, 1 equiv.) solubilized in 10 mL dichloroethane was added, and the mixture was stirred at 80 °C for 46 h. After cooling to room temperature, the reaction mixture was poured into an ice–water mixture and treated with aqueous NaOH (32%) until pH = 6 and then extracted with EtOAc. The organic phase was dried with Na_2_SO_4_ and concentrated *in vacuo* to obtain a crude mixture that was purified by column chromatography (EtOAc:cyclohexane 1:5 as eluent) to afford the pure product **34** (3.16 g, 70% yield).

10-methyl-10*H*-phenothiazine-3-carbaldehyde (**34**) has the same physico-chemical properties as described previously [37].

### 3.2. General Procedure for the Synthesis of Hydrazone Derivatives (***1**–**27***) 

A solution of aldehyde **28**–**39** (1 equiv.) and hydrazine derivative **40**–**49** (1 equiv.) in ethanol (5–10 mL) was stirred at reflux for 4–8 h. When the reaction was completed (TLC monitoring), the reaction mixture was cooled to room temperature, the product precipitated and was collected by filtration, washed with ethanol and purified by recrystallization from ethanol to obtain pure target hydrazone derivative as *E*-isomers (**1**–**27**).

#### 3.2.1. 1-(3-Bromophenyl)-2-(3,4,5-trimethoxybenzylidene)hydrazine (**1**)

The general procedure was used with 3,4,5-trimethoxybenzaldehyde (0.26 g, 1.3 mmol) and 3-bromophenylhydrazine (0.28 g, 1.3 mmol) to obtain pure compound **1** as a white solid (0.40 g, 1.0 mmol, 84% yield); mp (EtOH) 146–148 °C; Rf (EtOAc:Cyclohexane 1:1) = 0.63. ^1^H NMR (DMSO-*d_6_*, 500 MHz) δ ppm 3.88 (s, 3H, OC*H*_3_), 3.92 (s, 6H, 2OC*H*_3_), 6.97 (s, 2H, 2Ar*H*), 6.93–7.01 (m, 2H, 2Ar*H*), 7.11 (t, *J* = 8.0 Hz, 1H, Ar*H*), 7.29 (t, *J* = 2.0 Hz, 1H, Ar*H*), 7.58 (s, 1H, =C*H*), 7.67 (br s, 1H, N*H*). ^13^C NMR (DMSO-*d_6_*, 125 MHz) δ ppm 56.3 (2OCH_3_), 61.1 (OCH_3_), 103.5 (2CH), 111.5 (CH), 115.7 (CH), 122.9 (CH), 123.4 (C), 130.6 (C), 130.7 (CH), 138.5 (=CH), 138.9 (C), 145.9 (C), 153.6 (2C). IR ν (cm^−1^): 3245, 1618, 1625, 1585, 1455, 1379, 1357, 1278, 1098, 1072, 1043, 972, 878, 856, 827, 794, 733, 660. Elemental analysis calcd (%) for C_16_H_17_BrN_2_O_3_: C, 52.62; H, 4.69; N, 7.67; found: C, 52.89; H, 4.92; N, 7.88.

#### 3.2.2. 1-(3,4-Dimethylphenyl)-2-(4-nitrobenzylidene)hydrazine (**2**)

The general procedure was used with 4-nitrobenzaldehyde (0.20 g, 1.3 mmol) and 3,4-dimethylphenylhydrazine (0.22 g, 1.3 mmol) to obtain pure compound **2** as a dark red solid (0.23 g, 0.8 mmol, 65% yield); mp (EtOH) 118–120 °C; Rf (EtOAc:Cyclohexane 1:1) = 0.36. ^1^H NMR (CDCl_3_, 500 MHz) δ ppm 2.22 (s, 3H, C*H*_3_), 2.28 (s, 3H, C*H*_3_), 6.88 (d, *J* = 8.0 Hz, 1H, Ar*H*), 6.89 (s, 1H, Ar*H*), 7.06 (d, *J* = 8.0 Hz, 1H, Ar*H*), 7.64 (s, 1H, =C*H*), 7.75 (d, *J* = 8.5 Hz, 2H, 2Ar*H*), 7.91 (br s, 1H, N*H*), 8.21 (d, *J* = 8.5 Hz, 2H, 2Ar*H*). ^13^C NMR (CDCl_3_, 125 MHz) δ ppm 19.1 (CH_3_), 20.2 (CH_3_), 110.7 (CH), 114.6 (CH), 124.2 (2CH), 126.2 (2CH), 129.6 (C), 130.6 (CH), 133.2 (=CH), 137.9 (C), 141.7 (C), 142.1 (C), 146.9 (C). IR *ν* (cm^−1^): 3292, 1616, 1597, 1555, 1531, 1497, 1406, 1323, 1265, 1169, 1106, 912, 842, 812, 746, 690. Elemental analysis calcd (%) for C_15_H_15_N_3_O_2_: C, 66.90; H, 5.61; N, 15.60; found: C, 67.22; H, 5.90; N, 15.76.

#### 3.2.3. 2-(2-(4-Nitrobenzylidene)hydrazinyl)-4,5-dihydro-1*H*-imidazole (**3**)

The general procedure was used with 4-nitrobenzaldehyde (0.20 g, 1.3 mmol) and 2-hydrazinyl-4,5-dihydro-1*H*-imidazole (0.24 g, 1.3 mmol) to obtain pure compound **3** as a dark red solid (0.22 g, 0.9 mmol, 70% yield) with the same physico-chemical properties as described previously [38]; mp (EtOH) >250 °C; Rf (EtOAc:Cyclohexane 1:1) = 0.13. ^1^H NMR (DMSO-*d_6_*, 500 MHz) δ ppm 3.45 (br s, 4H, 2C*H*_2_), 6.87 (s, 1H, N*H*), 7.20 (s, 1H, N*H*), 7.91 (d, *J* = 8.5 Hz, 2H, 2Ar*H*), 8.05 (s, 1H, =C*H*), 8.17 (d, *J* = 8.5 Hz, 2H, 2Ar*H*). ^13^C NMR (DMSO-*d_6_*, 125 MHz) δ ppm 41.8 (CH_2_), 42.3 (CH_2_), 123.7 (2CH), 126.6 (2CH), 141.6 (=CH), 143.9 (C), 146.0 (C), 166.5 (C). IR *ν* (cm^−1^): 3109, 1622, 1590, 1539, 1495, 1481, 1398, 1375, 1320. 1288, 1167, 1103, 1055, 987, 923, 844, 777, 748, 690, 621. Elemental analysis calcd (%) for C_10_H_11_N_5_O_2_: C, 51.50; H, 4.75; N, 30.03; found: C, 51.74; H, 4.98; N, 30.29.

#### 3.2.4. 2-(2-(Thiophen-2-ylmethylene)hydrazinyl)-4,5-dihydro-1*H*-imidazole (**4**)

The general procedure was used with 2-thiophenecarboxaldehyde (0.20 g, 1.7 mmol) and 2-hydrazinyl-4,5-dihydro-1*H*-imidazole (0.32 g, 1.7 mmol) to obtain pure compound **4** as a white solid (0.26 g, 1.3 mmol, 74% yield) with the same physico-chemical properties as described previously [39]; mp (EtOH) >250 °C; Rf (EtOAc:Cyclohexane 1:1) = 0.1. ^1^H NMR (DMSO-*d_6_*, 500 MHz) δ ppm 3.70 (br s, 4H, 2C*H*_2_), 7.16 (t, *J* = 4.0 Hz, 1H, Ar*H*), 7.54 (d, *J* = 4.0 Hz, 1H, Ar*H*), 7.74 (d, *J* = 4.0 Hz, 1H, Ar*H*), 8.41 (s, 1H, =C*H*), 8.52 (br s, 1H, N*H*), 12.26 (br s, 1H, N*H*). ^13^C NMR (DMSO-*d_6_*, 125 MHz) δ ppm 42.8 (2CH_2_), 128.0 (CH), 130.1 (CH), 132.1 (CH), 137.5 (C), 143.5 (=CH), 157.5 (C). IR ν (cm^−1^): 3112, 1616, 1585, 1525, 1493, 1479, 1395, 1370, 1325, 1103, 1055, 985, 844, 748, 690, 621. Elemental analysis calcd (%) for C_8_H_10_N_4_S: C, 49.46; H, 5.19; N, 28.84; found: C, 49.76; H, 5.39; N, 29.01.

#### 3.2.5. 2-(2-(2,4-Dichlorobenzylidene)hydrazinyl)-4,5-dihydro-1*H*-imidazole (**5**)

The general procedure was used with 2,4-dichlorobenzaldehyde (0.30 g, 1.7 mmol) and 2-hydrazinyl-4,5-dihydro-1*H*-imidazole (0.31 g, 1.7 mmol) to obtain pure compound **5** as a white solid (0.30 g, 1.1 mmol, 68% yield); mp (EtOH) >250 °C; Rf (EtOAc:Cyclohexane 1:1) = 0.09. ^1^H NMR (DMSO-*d_6_*, 500 MHz) δ ppm 3.75 (br s, 4H, 2C*H*_2_), 7.56 (dd, *J* = 8.5, 2.0 Hz, 1H, Ar*H*), 7.74 (d, *J* = 2.0 Hz, 1H, Ar*H*), 7.24 (d, *J* = 8.5 Hz, 1H, Ar*H*), 8.54 (s, 1H, =C*H*), 8.91 (br s, 1H, N*H*), 12.61 (s, 1H, N*H*). ^13^C NMR (DMSO-*d_6_*, 125 MHz) δ ppm 42.8 (2CH_2_), 127.9 (CH), 128.6 (CH), 129.4 (CH), 129.7 (C), 134.1 (C), 135.8 (C), 142.9 (=CH), 157.6 (C). IR *ν* (cm^−1^): 3130, 1641, 1603, 1584, 1456, 1379, 1357, 1278, 1205, 1097, 1070, 1046, 997, 929, 877, 856, 827, 794, 736, 661. Elemental analysis calcd (%) for C_10_H_10_Cl_2_N_4_: C, 46.71; H, 3.92; N, 21.79; found: C, 47.02; H, 4.16; N, 22.04.

#### 3.2.6. 7-Chloro-4-(2-(3-chloro-4-methoxybenzylidene)hydrazinyl)quinoline (**6**)

The general procedure was used with 4-chloro-3-methoxybenzaldehyde (0.20 g, 1.1 mmol) and 7-chloro-4-hydrazinoquinoline (0.23 g, 1.1 mmol) to obtain pure compound **6** as a yellow solid (0.28 g, 0.8 mmol, 71% yield); mp (EtOH) 147–148 °C; Rf (EtOAc:Cyclohexane 1:1) = 0.31. ^1^H NMR (DMSO-*d_6_*, 500 MHz) δ ppm 3.92 (s, 3H, OC*H*_3_), 7.23 (d, *J* = 8.0 Hz, 1H, Ar*H*), 7.36–7.45 (m, 1H, Ar*H*), 7.54–7.63 (m, 1H, Ar*H*), 7.71 (dd, *J* = 8.0, 2.0 Hz, 1H, Ar*H*), 7.89 (d, *J* = 2.0 Hz, 2H, 2Ar*H*), 8.25–8.44 (m, 2H, 2Ar*H*), 8.58 (s, 1H, =C*H*), 11.24 (s, 1H, N*H*). ^13^C NMR (DMSO-*d_6_*, 125 MHz) δ ppm 56.3 (OCH_3_), 101.3 (CH), 113.0 (CH), 115.5 (C), 121.8 (C), 124.0 (CH), 124.9 (CH), 127.2 (CH), 127.4 (CH), 127.7 (CH), 128.4 (C), 133.8 (C), 141.8 (=CH), 147.0 (C), 149.2 (C), 152.1 (CH), 155.4 (C). IR ν (cm^−1^): 3232, 1616, 1572, 1534, 1440, 1420, 1366, 1192, 1025, 912, 850, 820, 752. Elemental analysis calcd (%) for C_17_H_13_Cl_2_N_3_O: C, 58.98; H, 3.78; N, 12.14; found: C, 59.25; H, 3.95; N, 12.37.

#### 3.2.7. 4-((2-(7-Chloroquinolin-4-yl)hydrazono)methyl)-2,6-dimethoxyphenol (**7**)

The general procedure was used with 3,5-dimethoxy-4-hydroxybenzaldehyde (0.30 g, 1.6 mmol) and 7-chloro-4-hydrazinoquinoline (0.31 g, 1.6 mmol) to obtain pure compound **7** as a yellow solid (0.59 g, 1.6 mmol, 69% yield); mp (EtOH) 220–222 °C; Rf (EtOAc:Cyclohexane 1:1) = 0.32. ^1^H NMR (DMSO-*d_6_*, 500 MHz) δ ppm 3.85 (s, 6H, 2OC*H*_3_), 7.06 (s, 2H, 2Ar*H*), 7.21–7.30 (m, 1H, Ar*H*), 7.31–7.60 (m, 1H, Ar*H*), 7.71–8.01 (m, 1H, Ar*H*), 8.28 (s, 1H, Ar*H*), 8.36 (d, *J* = 9.0 Hz, 1H, Ar*H*), 8.53 (br s, 1H, O*H*), 8.85 (s, 1H, =C*H*), 11.11 (br s, 1H, N*H*). ^13^C NMR (DMSO-*d_6_*, 125 MHz) δ ppm 56.1 (2OCH_3_), 101.0 (CH), 104.3 (2CH), 115.6 (C), 124.2 (CH), 124.6 (CH), 125.1 (2C), 127.6 (CH), 133.8 (C), 137.5 (C), 144.1 (=CH), 147.1 (C), 148.2 (2C), 152.0 (CH). IR *ν* (cm^−1^): 3292, 3213, 1616, 1574, 1502, 1447, 1416, 1366, 1302, 1194, 1094, 1026, 910, 858, 806, 756. Elemental analysis calcd (%) for C_18_H_16_ClN_3_O_3_: C, 60.42; H, 4.51; N, 11.74; found: C, 60.78; H, 4.69; N, 12.01.

#### 3.2.8. 3-((2-(7-Chloroquinolin-4-yl)hydrazono)methyl)-10-methyl-10*H*-phenothiazine (**8**)

The general procedure was used with 3-formyl-10-methylphenothiazine (0.25 g, 1 mmol) and 7-chloro-4-hydrazinoquinoline (0.20 g, 1 mmol) to obtain pure compound **8** as a yellow solid (0.34 g, 0.8 mmol, 79% yield); mp (EtOH) 239–241 °C; Rf (EtOAc:Cyclohexane 1:1) = 0.23. ^1^H NMR (DMSO-*d_6_*, 500 MHz) δ ppm 3.35 (s, 3H, NC*H*_3_), 6.96–7.20 (m, 3H, 3Ar*H*), 7.18 (d, *J* = 7.0 Hz, 1H, Ar*H*), 7.23 (t, *J* = 7.0 Hz, 1H, Ar*H*), 7.35–7.42 (m, 1H, Ar*H*), 7.53–7.61 (m, 3H, 3Ar*H*), 7.89 (s, 1H, Ar*H*), 8.27 (s, 1H, =C*H*), 8.35 (d, *J* = 9.0 Hz, 1H, Ar*H*), 8.53–8.61 (m, 1H, Ar*H*), 11.16 (s, 1H, N*H*). ^13^C NMR (DMSO-*d_6_*, 125 MHz) δ ppm 35.4 (NCH_3_), 101.3 (CH), 114.7 (CH), 114. 9 (CH), 115.6 (C), 121.6 (C), 122.7 (C), 122.9 (CH), 123.9 (CH), 124.3 (CH), 124.8 (CH), 126.8 (CH), 126.9 (CH), 127.7 (CH), 127.9 (CH), 129.2 (C), 133.8 (C), 142.4 (=CH), 144.7 (C), 146.2 (C), 147.0 (C), 149.2 (C), 152.0 (CH). IR *ν* (cm^−1^): 3404, 1618, 1570, 1460, 1427, 1354, 1331, 1248, 1209, 1127, 1089, 1049, 881, 812, 748, 638, 606. Elemental analysis calcd (%) for C_23_H_17_ClN_4_S: C, 66.26; H, 4.11; N, 13.44; found: C, 66.51; H, 4.28 N, 13.69.

#### 3.2.9. 3-((2-(4-Methoxyphenyl)hydrazono)methyl)-10-methyl-10*H*-phenothiazine (**9**)

The general procedure was used with 3-formyl-10-methylphenothiazine (0.25 g, 1 mmol) and 4-methoxyphenylhydrazine (0.18 g, 1 mmol) to obtain pure compound **9** as a yellow solid (0.30 g, 0.8 mmol, 81% yield); mp (EtOH) 188–190 °C; Rf (EtOAc:Cyclohexane 1:1) = 0.3. ^1^H NMR (DMSO-*d_6_*, 500 MHz) δ ppm 3.34 (s, 3H, NC*H*_3_), 3.69 (s, 3H, OC*H*_3_), 6.84 (d, *J* = 8 Hz, 2H, 2Ar*H*), 6.80–7.03 (m, 5H, 5Ar*H*), 7.18 (d, *J* = 7.5 Hz, 1H, Ar*H*), 7.24 (d, *J* = 7.5 Hz, 1H, Ar*H*), 7.38–7.47 (m, 2H, 2Ar*H*), 7.71 (s, 1H, =C*H*), 10.02 (s, 1H, N*H*). ^13^C NMR (DMSO-*d_6_*, 125 MHz) δ ppm 35.3 (NCH_3_), 55.3 (OCH_3_), 112.9 (2CH), 114.6 (2CH), 114.7 (CH), 116.8 (CH), 121.7 (C), 122.5 (C), 122.6 (CH), 123.1 (CH), 125.2 (CH), 126.8 (CH), 127.8 (CH), 130.7 (C), 134.4 (=CH), 139.5 (C), 144.6 (C), 145.0 (C), 152.5 (C). IR *ν* (cm^−1^): 3281, 1683, 1601, 1584, 1522, 1501, 1466, 1448, 1352, 1331, 1259, 1243, 1142, 1034, 911, 827, 765, 609. Elemental analysis calcd (%) for C_21_H_19_N_3_OS: C, 66.26; H, 4.11; N, 13.44; found: C, 66.50; H, 4.33; N, 13.69.

#### 3.2.10. 3-((2-(Benzo[*d*]thiazol-2-yl)hydrazono)methyl)-10-methyl-10*H*-phenothiazine (**10**)

The general procedure was used with 3-formyl-10-methylphenothiazine (0.30 g, 1.3 mmol) and 2-hydrazinobenzothiazole (0.20 g, 1.3 mmol) to obtain pure compound **10** as a yellow solid (0.38 g, 0.9 mmol, 81% yield); mp (EtOH) >250 °C; Rf (EtOAc:Cyclohexane 1:1) = 0.36. ^1^H NMR (DMSO-*d_6_*, 500 MHz) δ ppm 3.51 (s, 3H, NC*H*_3_), 6.96–7.05 (m, 3H, 3Ar*H*), 7.10 (t, *J* = 7.5 Hz, 1H, Ar*H*), 7.19 (d, *J* = 7.5 Hz, 1H, Ar*H*), 7.24 (t, *J* = 7.0 Hz, 1H, Ar*H*), 7.29 (t, *J* = 7.0 Hz, 1H, Ar*H*), 7.35–7.54 (m, 3H, Ar*H*), 7.70–7.82 (m, 1H, Ar*H*), 8.03 (s, 1H, =C*H*), 12.23 (br s, 1H, N*H*). ^13^C NMR (DMSO-*d_6_*, 125 MHz) δ ppm 35.6 (NCH_3_), 114.8 (CH), 114.9 (2CH), 121.5 (CH), 121.6 (C), 122.6 (C), 122.9 (2CH), 124.1 (CH), 126.0 (CH), 126.7 (CH), 126.9 (2CH), 128.0 (=CH), 128.9 (C), 144.6 (2C), 146.3 (2C), 167.0 (C). IR *ν* (cm^−1^): 3090, 1624, 1578, 1464, 1439, 1335, 1254, 1219, 1113, 923, 883, 800, 736, 715. Elemental analysis calcd (%) for C_21_H_16_N_4_S_2_: C, 64.92; H, 4.15; N, 14.42; found: C, 64.78; H, 4.03; N, 14.23.

#### 3.2.11. 3-((2-(4-Bromophenyl)hydrazono)methyl)-10-methyl-10*H*-phenothiazine (**11**)

The general procedure was used with 3-formyl-10-methylphenothiazine (0.30 g, 1.3 mmol) and 4-bromophenylhydrazine (0.28 g, 1.3 mmol) to obtain pure compound **11** as a yellow solid (0.38 g, 0.9 mmol, 75% yield); mp (EtOH) 203–204 °C; Rf (EtOAc:Cyclohexane 1:1) = 0.32. ^1^H NMR (DMSO-*d_6_*, 500 MHz) δ ppm 3.34 (s, 3H, NC*H_3_*), 6.93–7.02 (m, 5H, 5Ar*H*), 7.18 (d, *J* = 8.0 Hz, 1H, ArH), 7.23 (td, *J* = 8.0, 2.0 Hz, 1H, Ar*H*), 7.34 (d, *J* = 8.0 Hz, 2H, 2Ar*H*), 7.42–7.48 (m, 2H, 2Ar*H*), 7.77 (s, 1H, =C*H*), 10.39 (s, 1H, N*H*). ^13^C NMR (DMSO-*d_6_*, 125 MHz) δ ppm 35.3 (NCH_3_), 109.3 (C), 113.9 (2CH), 114.7 (CH), 114.8 (CH), 121.6 (C), 122.6 (C), 122.7 (CH), 123.5 (CH), 125.7 (CH), 126.9 (CH), 127.9 (CH), 130.2 (C), 131.7 (2CH), 136.5 (=CH), 144.7 (C), 144.9 (C), 145.2 (C). IR *ν* (cm^−1^): 3273, 1683, 1548, 1453, 1331, 1257, 1331, 1257, 1139, 1068, 912, 831, 810, 756, 601. Elemental analysis calcd (%) for C_20_H_16_BrN_3_S: C, 58.54; H, 3.93; N, 10.24; found: C, 58.72; H, 4.16; N, 10.44.

#### 3.2.12. 3-((2-(3-Bromophenyl)hydrazono)methyl)-10-methyl-10*H*-phenothiazine (**12**)

The general procedure was used with 3-formyl-10-methylphenothiazine (0.30 g, 1.3 mmol) and 3-bromophenylhydrazine (0.28 g, 1.3 mmol) to obtain pure compound **12** as a yellow solid (0.36 g, 0.8 mmol, 72% yield); mp (EtOH) 128–130 °C; Rf (EtOAc:Cyclohexane 1:1) = 0.3. ^1^H NMR (CDCl_3_, 500 MHz) δ ppm 3.37 (s, 3H, NC*H*_3_), 6.76 (d, *J* = 8.0 Hz, 1H, Ar*H*), 6.82 (d, *J* = 8.0 Hz, 1H, Ar*H*), 6.90- 6.98 (m, 3H, 3Ar*H*), 7.10 (t, *J* = 8.0 Hz, 1H, Ar*H*), 7.14- 7.21 (m, 2H, 2Ar*H*), 7.29 (s, 1H, Ar*H*), 7.38 (dd, *J* = 8.2 Hz, 1H, Ar*H*), 7.45 (d, *J* = 2.0 Hz, 1H, Ar*H*), 7.47–7.55 (m, 2H, =C*H*+ N*H*). ^13^C NMR (CDCl_3_, 125 MHz) δ ppm 35.6 (NCH_3_), 111.4 (CH), 114.1 (CH), 114.4 (CH), 115.6 (CH), 122.8 (CH), 122.9 (CH), 123.0 (C), 123.4 (C), 123.9 (C), 124.7 (CH), 126.1 (CH), 127.4 (CH), 127.7 (CH), 129.6 (C), 130.7 (CH), 137.7 (=CH), 145.4 (C), 146.1 (C), 146.3 (C). IR *ν* (cm^−1^): 3297, 1686, 1586, 1464, 1336, 1257, 1236, 1257, 1236, 1123, 1082, 1063, 987, 916, 847, 810, 746, 680. Elemental analysis calcd (%) for C_20_H_16_BrN_3_S: C, 58.54; H, 3.93; N, 10.24; found: C, 58.36; H, 3.87; N, 10.15.

#### 3.2.13. 4-(2-((1*H*-Indol-3-yl)methylene)hydrazinyl)-7-chloroquinoline (**13**)

The general procedure was used with indole-3-carboxaldehyde (0.20 g, 1.3 mmol) and 7-chloro-4-hydrazinoquinoline (0.26 g, 1.3 mmol) to obtain pure compound **13** as a yellow solid (0.32 g, 0.9 mmol, 73% yield) with the same physico-chemical properties as described previously [28]; mp (EtOH) >250 °C; Rf (EtOAc:Cyclohexane 1:1) = 0. ^1^H NMR (DMSO-*d_6_*, 500 MHz) δ ppm 7.21–7.28 (m, 2H, 2Ar*H*), 7.35 (d, *J* = 4.5 Hz, 1H, Ar*H*), 7.43–7.51 (m, 1H, Ar*H*), 7.55 (d, *J* = 8.5 Hz, 1H, Ar*H*), 7.83–7.91 (m, 2H, 2Ar*H*), 8.30–8.35 (m, 1H, Ar*H*), 8.40 (d, *J* = 8.5 Hz, 1H, Ar*H*), 8.60 (d, *J* = 4.5 Hz, 1H, Ar*H*), 8.63 (s, 1H, =C*H*), 10.94 (s, 1H, N*H*), 11.61 (s, 1H, N*H*). ^13^C NMR (DMSO-*d_6_*, 125 MHz) δ ppm 100.3 (CH), 112.0 (CH), 115.6 (C), 120.6 (CH), 121.7 (CH), 122.6 (CH), 124.1 (CH), 124.2 (C), 124.4 (CH), 127.6 (CH), 130.0 (CH), 133.6 (C), 137.2 (2C), 141.5 (=CH), 147.2 (C), 149.3 (C), 152.1 (CH). IR *ν* (cm^−1^): 3332, 1614, 1572, 1483, 1447, 1416, 1356, 1317, 1277, 1244, 1198, 1123, 1080, 1022, 860, 810, 747. Elemental analysis calcd (%) for C_18_H_13_ClN_4_: C, 67.40; H, 4.08; N, 17.47; found: C, 67.72; H, 4.35; N, 17.39.

#### 3.2.14. 3-((2-(6-Chloropyridin-2-yl)hydrazono)methyl)-1*H*-indole (**14**)

The general procedure was used with indole-3-carboxaldehyde (0.25 g, 1.7 mmol) and 2-chloro-6-hydrazinopyridine (0.25 g, 1.7 mmol) to obtain pure compound **14** as a beige solid (0.38 g, 1.4 mmol, 82% yield); mp (EtOH) 206–208 °C; Rf (EtOAc:Cyclohexane 1:1) = 0.32. ^1^H NMR (DMSO-*d_6_*, 500 MHz) δ ppm 6.71 (d, *J* = 7.5 Hz, 1H, Ar*H*), 7.12–7.23 (m, 3H, 3Ar*H*), 7.43 (d, *J* = 7.5 Hz, 1H, Ar*H*), 7.68 (t, *J* = 7.5 Hz, 1H, Ar*H*), 7.72 (d, *J* = 2.0 Hz, 1H, Ar*H*), 8.22 (d, *J* = 7.5 Hz, 1H, Ar*H*), 8.25 (s, 1H, =C*H*), 10.89 (s, 1H, N*H*), 11.46 (s, 1H, N*H*). ^13^C NMR (DMSO-*d_6_*, 125 MHz) δ ppm 104.1 (CH), 111.8 (CH), 112.1 (C), 112.3 (CH), 120.3 (CH), 121.6 (CH), 122.5 (CH), 124.1 (C), 129.0 (CH), 137.1 (C), 138.6 (=CH), 141.0 (CH), 148.2 (C), 157.5 (C). IR *ν* (cm^−1^): 3375, 3273, 1612, 1587, 1639, 1508, 1454, 1418, 1311, 1242, 1089, 1072, 975, 929, 771, 739, 704, 630. Elemental analysis calcd (%) for C_14_H_11_ClN_4_: C, 62.11; H, 4.10; N, 20.70; found: C, 62.37; H, 4.32; N, 20.96.

#### 3.2.15. 3-((2-(5-Bromopyridin-2-yl)hydrazono)methyl)-1*H*-indole (**15**)

The general procedure was used with indole-3-carboxaldehyde (0.25 g, 1.7 mmol) and 5-bromo-2-hydrazinopyridine (0.33 g, 1.7 mmol) to obtain pure compound **15** as a beige solid (0.54 g, 1.7 mmol, 79% yield); mp (EtOH) 244–246 °C; Rf (EtOAc:Cyclohexane 1:1) = 0.56. ^1^H NMR (DMSO-*d_6_*, 500 MHz) δ ppm 7.12–7.23 (m, 3H, 3Ar*H*), 7.43 (d, *J* = 7.5 Hz, 1H, Ar*H*), 7.69 (d, *J* = 2.5 Hz, 1H, Ar*H*), 7.82 (dd, *J* = 7.5, 2.5 Hz, 1H, Ar*H*), 8.15 (d, *J* = 2.5 Hz, 1H, Ar*H*), 8.20 (d, *J* = 7.5 Hz, 1H, Ar*H*), 8.26 (s, 1H, =C*H*), 10.69 (s, 1H, N*H*), 11.44 (s, 1H, N*H*). ^13^C NMR (DMSO-*d_6_*, 125 MHz) δ ppm 106.8 (C), 107.5 (CH), 111.8 (CH), 112.2 (C), 120.3 (CH), 121.6 (CH), 122.4 (CH), 124.1 (C), 128.7 (CH), 137.1 (C), 138.1 (=CH), 140.2 (CH), 148.0 (CH), 156.3 (C). IR *ν* (cm^−1^): 3378, 3180, 1616, 1585, 1537, 1445, 1386, 1362, 1304, 1243, 1130, 1083, 999, 808, 785, 743, 642, 619. Elemental analysis calcd (%) for C_14_H_11_BrN_4_: C, 53.35; H, 3.52; N, 17.78; found: C, 53.50; H, 3.77; N, 17.95.

#### 3.2.16. 7-Chloro-4-(2-((5-methoxy-1*H*-indol-3-yl)methylene)hydrazinyl)quinoline (**16**)

The general procedure was used with 5-methoxyindole-3-carboxaldehyde (0.30 g, 1.7 mmol) and 7-chloro-4-hydrazinoquinoline (0.33 g, 1.7 mmol) to obtain pure compound **16** as a yellow solid (0.46 g, 1.3 mmol, 78% yield) with the same physico-chemical properties as described previously [40]; mp (EtOH) > 255 °C; Rf (EtOAc:Cyclohexane 1:1) = 0.32. ^1^H NMR (DMSO-*d_6_*, 500 MHz) δ ppm 3.88 (s, 3H, OC*H*_3_), 6.89 (dd, *J* = 8.5, 2.0 Hz, 1H, Ar*H*), 7.29 (d, *J* = 5.0 Hz, 1H, Ar*H*), 7.37 (d, *J* = 8.5 Hz, 1H, Ar*H*), 7.55 (d, *J* = 2.0 Hz, 1H, Ar*H*), 7.72–7.95 (m, 3H, 3Ar*H*), 8.39 (d, *J* = 5.0 Hz, 1H, Ar*H*), 8.58 (d, *J* = 5.0 Hz, 1H, Ar*H*), 8.61 (s, 1H, =C*H*), 10.92 (s, 1H, N*H*), 11.47 (s, 1H, N*H*). ^13^C NMR (DMSO-*d_6_*, 125 MHz) δ ppm 55.2 (OCH_3_), 100.0 (CH), 103.6 (CH), 111.8 (C), 112.2 (CH), 112.6 (CH), 115.6 (C), 124.2 (CH), 124.8 (CH), 127.5 (CH), 130.3 (CH), 132.1 (2C), 133.7 (C), 141.6 (=CH), 147.2 (C), 149.3 (C), 152.0 (CH), 154.5 (C). IR *ν* (cm^−1^): 3355, 3235, 1616, 1581, 1429, 1423, 1369, 1292, 1209, 1123, 1076, 1028, 921, 873, 839, 794, 761, 721, 621. Elemental analysis calcd (%) for C_19_H_15_ClN_4_O: C, 65.05; H, 4.31; N, 15.97; found: C, 65.31; H, 4.59; N, 14.98.

#### 3.2.17. 2-(2-((5-Methoxy-1*H*-indol-3-yl)methylene)hydrazinyl)benzo[*d*]thiazole (**17**)

The general procedure was used with 5-methoxyindole-3-carboxaldehyde (0.30 g, 1.7 mmol) and 2-hydrazinobenzothiazole (0.28 g, 1.7 mmol) to obtain pure compound **17** as a yellow solid (0.32 g, 0.9 mmol, 58% yield); mp (EtOH) >250 °C; Rf (EtOAc:Cyclohexane 1:1) = 0.04. ^1^H NMR (DMSO-*d_6_*, 500 MHz) δ ppm 3.91 (s, 3H, OC*H*_3_), 6.88 (dd, *J* = 8.5, 2.0 Hz, 1H, Ar*H*), 7.06 (t, *J* = 7.5 Hz, 1H, Ar*H*), 7.27 (t, *J* = 7.5 Hz, 1H, Ar*H*), 7.35 (d, *J* = 8.5 Hz, 1H, Ar*H*), 7.39 (d, *J* = 2.0 Hz, 1H, Ar*H*), 7.75 (d, *J* = 2.0 Hz, 1H, Ar*H*), 7.77 (d, *J* = 7.5 Hz, 1H, Ar*H*), 7.83 (d, *J* = 2.0 Hz, 1H, Ar*H*), 8.33 (s, 1H, =C*H*), 11.43 (s, 1H, N*H*), 11.93 (br s, 1H, N*H*). ^13^C NMR (DMSO-*d_6_*, 125 MHz) δ ppm 55.4 (OCH_3_), 103.7 (CH), 111.4 (C), 112.4 (2CH), 112.6 (CH), 117, 4 (C), 121.0 (CH), 121.5 (CH), 124.7 (C), 125.9 (CH), 129.1 (C), 130.3 (CH), 132.0 (C), 141.8 (=CH), 154.5 (C), 166.5 (C). IR *ν* (cm^−1^): 3427, 1614, 1576, 1485, 1440, 1418, 1261, 1207, 1121, 1096, 1022, 921, 889, 858, 810, 796, 738, 715. Elemental analysis calcd (%) for C_17_H_14_N_4_OS: C, 63.33; H, 4.38; N, 17.38; found: C, 63.38; H, 4.26; N, 17.17.

#### 3.2.18. 3-((2-(5-Bromopyridin-2-yl)hydrazono)methyl)-5-methoxy-1*H*-indole (**18**)

The general procedure was used with 5-methoxyindole-3-carboxaldehyde (0.25 g, 1.4 mmol) and 5-bromo-2-hydrazinopyridine (0.27 g, 1.4 mmol) to obtain pure compound **18** as a white solid (0.32 g, 0.9 mmol, 65% yield); mp (EtOH) 212–214 °C; Rf (EtOAc:Cyclohexane 1:1) = 0.36. ^1^H NMR (DMSO-*d_6_*, 500 MHz) δ ppm 3.83 (s, 3H, OC*H*_3_), 6.85 (dd, *J* = 9.0, 2.5 Hz, 1H, Ar*H*), 7.13 (d, *J* = 9.0 Hz, 1H, Ar*H*), 7.33 (d, *J* = 9.0 Hz, 1H, Ar*H*), 7.65 (d, *J* = 2.5 Hz, 1H, Ar*H*), 7.71 (d, *J* = 2.5 Hz, 1H, Ar*H*), 7.84 (dd, *J* = 9.0, 2.5 Hz, 1H, Ar*H*), 8.15 (d, *J* = 2.5 Hz, 1H, Ar*H*), 8.25 (s, 1H, =C*H*), 10.68 (s, 1H, N*H*), 11.31 (s, 1H, N*H*). ^13^C NMR (DMSO-*d_6_*, 125 MHz) δ ppm 55.5 (OCH_3_), 103.5 (CH), 106.7 (C), 107.3 (CH), 111.9 (C), 112.0 (CH), 112.4 (CH), 124.6 (C), 129.0 (CH), 132.0 (C), 138.2 (=CH), 140.2 (CH), 148.1 (CH), 154.2 (C), 156.3 (C). IR *ν* (cm^−1^): 3321, 3217, 1618, 1588, 1537, 1489, 1436, 1381, 1300, 1258, 1211, 1107, 1076, 1026, 993, 921, 846, 776, 729, 611. Elemental analysis calcd (%) for C_15_H_13_BrN_4_O: C, 52.19; H, 3.80; N, 16.23; found: C, 52.50; H, 4.06; N, 16.42.

#### 3.2.19. 3-((2-(6-Chloropyridin-2-yl)hydrazono)methyl)-5-methoxy-1*H*-indole (**19**)

The general procedure was used with 5-methoxyindole-3-carboxaldehyde (0.25 g, 1.4 mmol) and 2-chloro-6-hydrazinopyridine (0.21 g, 1.4 mmol) to obtain pure compound **19** as a white solid (0.31 g, 1.0 mmol, 72% yield); mp (EtOH) 200–202 °C; Rf (EtOAc:Cyclohexane 1:1) = 0.32. ^1^H NMR (DMSO-*d_6_*, 500 MHz) δ ppm 3.84 (s, 3H, OC*H*_3_), 6.71 (d, *J* = 8.5 Hz, 1H, Ar*H*), 6.85 (dd, *J* = 8.5, 2.0 Hz, 1H, Ar*H*), 7.09 (d, *J* = 8.5 Hz, 1H, Ar*H*), 7.33 (d, *J* = 8.5 Hz, 1H, Ar*H*), 7.64–7.71 (m, 2H, 2Ar*H*), 7.73 (d, *J* = 2.0 Hz, 1H, Ar*H*), 8.23 (s, 1H, =C*H*), 10.87 (s, 1H, N*H*), 11.33 (s, 1H, N*H*). ^13^C NMR (DMSO-*d_6_*, 125 MHz) δ ppm 55.2 (OCH_3_), 103.5 (CH), 103.8 (CH), 111.8 (C), 112.1 (CH), 112.2 (CH), 112.5 (CH), 124.6 (C), 129.3 (CH), 132.0 (C), 138.8 (=CH), 141.0 (CH), 148.3 (C), 154.3 (C), 157.5 (C). IR *ν* (cm^−1^): 3420, 3222, 1620, 1593, 1556, 1483, 1428, 1306, 1256, 12071093, 1067, 1022, 979, 912, 844, 802, 763, 698, 616. Elemental analysis calcd (%) for C_15_H_13_ClN_4_O: C, 59.91; H, 4.36; N, 18.63; found: C, 60.18; H, 4.44; N, 18.79.

#### 3.2.20. 7-Chloro-4-(2-((1-methyl-1*H*-indol-3-yl)methylene)hydrazinyl)quinoline (**20**)

The general procedure was used with 1-methyl-1*H*-indole-3-carbaldehyde (0.25 g, 1.6 mmol) and 7-chloro-4-hydrazinoquinoline (0.30 g, 1.6 mmol) to obtain pure compound **20** as a yellow solid (0.36 g, 1.0 mmol, 68% yield); mp (EtOH) >250 °C; Rf (EtOAc:Cyclohexane 1:1) = 0.32. ^1^H NMR (DMSO-*d_6_*, 500 MHz) δ ppm 3.84 (s, 3H, NC*H*_3_), 7.22–7.37 (m, 3H, 3Ar*H*), 7.46–7.61 (m, 2H, 2Ar*H*), 7.81–7.91 (m, 2H, 2Ar*H*), 8.32 (d, *J* = 8.0 Hz, 1H, Ar*H*), 8.38 (d, *J* = 8 Hz, 1H, Ar*H*), 8.59 (s, 2H, =C*H* + Ar*H*), 10.92 (s, 1H, N*H*). ^13^C NMR (DMSO-*d_6_*, 125 MHz) δ ppm 32.8 (NCH_3_),100.3 (CH), 110.3 (CH), 111.0 (C), 115.5 (C), 120.8 (CH), 121.9 (CH), 122.7 (CH), 124.0 (CH), 124.4 (CH), 124.5 (C), 127.6 (CH), 133.6 (CH), 138.7 (2C), 140.9 (=CH), 147.1 (C), 149.3 (C), 152.1 (CH). IR *ν* (cm^−1^): 3216, 1618, 1570, 1493, 1421, 1371, 1248, 1200, 1124, 1078, 862, 814, 739, 617. Elemental analysis calcd (%) for C_19_H_15_ClN_4_: C, 68.16; H, 4.52; N, 16.73; found: C, 68.34; H, 4.60; N, 16.91.

#### 3.2.21. 3-((2-(5-Bromopyridin-2-yl)hydrazono)methyl)-1-methyl-1*H*-indole (**21**)

The general procedure was used with 1-methyl-1*H*-indole-3-carbaldehyde (0.25 g, 1.6 mmol) and 5-bromo-2-hydrazinopyridine (0.29 g, 1.6 mmol) to obtain pure compound **21** as a beige solid (0.28 g, 0.8 mmol, 54% yield); mp (EtOH) 204–206 °C; Rf (EtOAc:Cyclohexane 1:1) = 0.3. ^1^H NMR (CDCl_3_, 500 MHz) δ ppm 3.81 (s, 3H, C*H*_3_), 7.24 (s, 1H, Ar*H*), 7.27–7.36 (m, 4H, 4Ar*H*), 7.70 (dd, *J* = 8.0, 2.0 Hz, 1H, Ar*H*), 7.97 (s, 1H, =C*H*), 8.16 (d, *J* = 2.0 Hz, 1H, Ar*H*), 8.24 (br s, 1H, N*H*), 8.32 (d, *J* = 8.0 Hz, 1H, Ar*H*). ^13^C NMR (CDCl_3_, 125 MHz) δ ppm 33.3 (CH_3_), 108.8 (CH), 108.9 (C), 109.6 (CH), 111.9 (C), 121.1 (CH), 122.4 (CH), 123.3 (CH), 125.2 (C), 131.0 (CH), 136.7 (CH), 137.9 (C), 140.5 (=CH), 148.3 (CH), 155.9 (C). IR *ν* (cm^−1^): 3190, 1622, 1586, 1537, 1444, 1375, 1302, 1124, 1074, 996, 919, 815, 740, 686, 634. Elemental analysis calcd (%) for C_15_H_13_BrN_4_: C, 54.73; H, 3.98; N, 17.02; found: C, 54.59; H, 3.86; N, 16.94.

#### 3.2.22. 3-((2-(6-Chloropyridin-2-yl)hydrazono)methyl)-1-methyl-1*H*-indole (**22**)

The general procedure was used with 1-methyl-1*H*-indole-3-carbaldehyde (0.25 g, 1.6 mmol) and 2-chloro-6-hydrazinopyridine (0.23 g, 1.6 mmol) to obtain pure compound **22** as a beige solid (0.34 g, 1.1 mmol, 76% yield); mp (EtOH) 205–207 °C; Rf (EtOAc:Cyclohexane 1:1) = 0.3. ^1^H NMR (CDCl_3_, 500 MHz) δ ppm 3.80 (s, 3H, NC*H*_3_), 6.73 (d, *J* = 7.0 Hz, 1H, Ar*H*), 7.23 (s, 1H, ArH), 7.25–7.37 (m, 4H, 4Ar*H*), 7.56 (t, *J* = 8.0 Hz, 1H, Ar*H*), 7.94 (s, 1H, =C*H*), 8.25 (s, 1H, N*H*), 8.33 (d, *J* = 8.0 Hz, 1H, Ar*H*). ^13^C NMR (CDCl_3_, 125 MHz) δ ppm 33.2 (NCH_3_), 105.2 (CH), 109.6 (CH), 111.8 (C), 114.0 (CH), 121.1 (CH), 122.4 (CH), 123.2 (CH), 125.2 (C), 131.2 (CH), 137.2 (=CH), 137.9 (C), 140.5 (CH), 149.0 (C), 157.1 (C). IR *ν* (cm^−1^): 3307, 1618, 1586, 1553, 1472, 1425, 1312, 1192, 1091, 977, 933, 773, 749, 698, 619. Elemental analysis calcd (%) for C_15_H_13_ClN_4_: C, 63.27; H, 4.60; N, 19.68; found: C, 63.47; H, 4.91; N, 19.95.

#### 3.2.23. 7-Chloro-4-(2-((5-methoxy-1-methyl-1H-indol-3-yl)methylene)hydrazinyl)quinoline (**23**)

The general procedure was used with 5-bromo-1-methyl-1*H*-indole-3-carbaldehyde (0.40 g, 2.1 mmol) and 7-chloro-4-hydrazinoquinoline (0.40 g, 2.1 mmol) to obtain pure compound **23** as a yellow solid (0.52 g, 1.4 mmol, 68% yield); mp (EtOH) 221–223 °C; Rf (EtOAc:Cyclohexane 1:1) = 0. ^1^H NMR (DMSO-*d_6_*, 500 MHz) δ ppm 3.81 (s, 3H, NC*H*_3_), 3.89 (s, 3H, OC*H*_3_), 6.94 (d, *J* = 8.0 Hz, 1H, Ar*H*), 7.28 (d, *J* = 5.0 Hz, 1H, Ar*H*), 7.44 (d, *J* = 8.0 Hz, 1H, Ar*H*), 7.55 (d, *J* = 8.5 Hz, 1H, Ar*H*), 7.78- 7.85 (m, 2H, 2Ar*H*), 7.86 (s, 1H, Ar*H*), 8.38 (d, *J* = 8.5 Hz, 1H, Ar*H*), 8.53–8.64 (m, 2H, Ar*H* + =C*H*), 10.92 (s, 1H, N*H*). ^13^C NMR (DMSO-*d_6_*, 125 MHz) δ ppm 33.0 (NCH_3_), 55.3 (OCH_3_), 100.1 (CH), 103.7 (CH), 110.5 (C), 111.2 (CH), 112.2 (CH), 115.6 (C), 124.0 (CH), 124.5 (CH), 125.1 (C), 127.6 (CH), 132.8 (CH), 133.6 (C), 133.8 (C), 141.1 (=CH), 147.1 (C), 149.3 (C), 152.1 (CH), 154.7 (C). IR *ν* (cm^−1^): 3168, 1612, 1565, 1527, 1425, 1372, 1360, 1305, 1251, 1115, 854, 784, 637, 604. Elemental analysis calcd (%) for C_20_H_17_ClN_4_O: C, 65.84; H, 4.70; N, 15.36; found: C, 65.67; H, 4.51; N, 15.22.

#### 3.2.24. 2-(2-((5-Methoxy-1-methyl-1*H*-indol-3-yl)methylene)hydrazinyl)quinoline (**24**)

The general procedure was used with 5-bromo-1-methyl-1*H*-indole-3-carbaldehyde (0.30 g, 1.6 mmol) and 2-hydrazinoquinoline (0.25 g, 1.6 mmol) to obtain pure compound **24** as a white solid (0.37 g, 1.1 mmol, 70% yield); mp (EtOH) 171–173 °C; Rf (EtOAc:Cyclohexane 1:1) = 0.23. ^1^H NMR (DMSO-*d_6_*, 500 MHz) δ ppm 3.78 (s, 3H, C*H*_3_), 3.89 (s, 3H, OC*H*_3_), 6.92 (d, *J* = 8.0 Hz, 1H, Ar*H*), 7.17–7.31 (m, 1H, Ar*H*), 7.40 (d, *J* = 8.0 Hz, 1H, Ar*H*), 7.47–7.62 (m, 3H, 3Ar*H*), 7.66–7.70 (m, 1H, Ar*H*), 7.73 (d, *J* = 8.0 Hz, 1H, Ar*H*), 7.82 (s, 1H, Ar*H*), 8.20 (d, *J* = 8.0 Hz, 1H, Ar*H*), 8.26 (s, 1H, =C*H*), 10.99 (s, 1H, N*H*). ^13^C NMR (DMSO-*d_6_*, 125 MHz) δ ppm 32.9 (CH_3_), 55.2 (OCH_3_), 103.6 (CH), 109.0 (CH), 111.0 (CH), 112.1 (CH), 121.9 (CH), 123.8 (C), 125.0 (CH), 125.5 (C), 127.8 (CH), 129.6 (CH), 132.6 (CH), 132.7 (2C), 137.3 (CH), 137.9 (CH), 147.5 (C), 154.5 (C), 156.0 (C). IR *ν* (cm^−1^): 3309, 1655, 1604, 1568, 1541, 1504, 1430, 1345, 1256, 1114, 1040, 931, 862, 815, 788, 763, 678, 643. Elemental analysis calcd (%) for C_20_H_18_N_4_O: C, 72.71; H, 5.49; N, 16.96; found: C, 73.04; H, 5.71; N, 17.13.

#### 3.2.25. 4-(2-((5-Bromo-1*H*-indol-3-yl)methylene)hydrazinyl)-7-chloroquinoline (**25**)

The general procedure was used with 5-bromoindole-3-carboxaldehyde (0.25 g, 1.1 mmol) and 7-chloro-4-hydrazinoquinoline (0.22 g, 1.1 mmol) to obtain pure compound **25** as a yellow solid (0.32 g, 0.8 mmol, 72% yield); mp (EtOH) 247–249 °C; Rf (EtOAc:Cyclohexane 1:1) = 0.3. ^1^H NMR (DMSO-*d_6_*, 500 MHz) δ ppm 7.20–7.81 (m, 4H, 4Ar*H*), 7.85–8.20 (m, 2H, 2Ar*H*), 8.21–8.50 (m, 2H, 2Ar*H*), 8.60 (s, 2H, =C*H* + Ar*H*), 10.97 (s, 1H, N*H*), 11.80 (s, 1H, N*H*). ^13^C NMR (DMSO-*d_6_*, 125 MHz) δ ppm 100.1 (CH), 111.5 (C), 113.1 (C), 114.0 (CH), 115.5 (C), 123.7 (CH), 124.0 (CH), 124.5 (CH), 125.1 (CH), 125.8 (C), 127.6 (CH), 131.2 (CH), 133.7 (C), 135.9 (C), 141.0 (=CH), 147.1 (C), 149.3 (C), 152.0 (CH). IR *ν* (cm^−1^): 3348, 3033, 1614, 1580, 1483, 1447, 1420, 1368, 1291, 1236, 1200, 1135, 1078, 1022, 929, 854, 798, 669, 622. Elemental analysis calcd (%) for C_18_H_12_BrClN_4_: C, 54.09; H, 3.03; N, 14.02; found: C, 54.34; H, 3.23; N, 14.19.

#### 3.2.26. 5-Bromo-3-((2-(5-bromopyridin-2-yl)hydrazono)methyl)-1*H*-indole (**26**)

The general procedure was used with 5-bromoindole-3-carboxaldehyde (0.25 g, 1.1 mmol) and 5-bromo-2-hydrazinopyridine (0.20 g, 1.1 mmol) to obtain pure compound **26** as a beige solid (0.31 g, 0.7 mmol, 71% yield); mp (EtOH) 209–211 °C; Rf (EtOAc:Cyclohexane 1:1) = 0.23. ^1^H NMR (DMSO-*d_6_*, 500 MHz) δ ppm 7.06 (d, *J* = 9.0 Hz, 1H, Ar*H*), 7.32 (dd, *J* = 8.5, 2.5 Hz, 1H, Ar*H*), 7.41 (d, *J* = 8.5 Hz, 1H, Ar*H*), 7.78 (d, *J* = 3.0 Hz, 1H, Ar*H*), 7.87 (dd, *J* = 9.0, 2.5 Hz, 1H, Ar*H*), 8.16 (d, *J* = 2.5 Hz, 1H, Ar*H*), 8.24 (s, 1H, =C*H*), 8.30 (d, *J* = 3.0 Hz, 1H, Ar*H*) 10.75 (s, 1H, N*H*), 11.65 (s, 1H, N*H*). ^13^C NMR (DMSO-*d_6_*, 125 MHz) δ ppm 107.1 (CH), 107.2 (C), 111.8 (C), 112.8 (C), 113.9 (CH), 123.6 (CH), 124.9 (CH), 125.8 (C), 130.0 (CH), 135.7 (C), 137.7 (=CH), 140.3 (CH), 148.2 (CH), 156.1 (C). IR *ν* (cm^−1^): 3406, 3192, 1620, 1590, 1543, 1514, 1444, 1425,1383, 1290, 1134, 1086, 997, 873, 819, 786, 664. Elemental analysis calcd (%) for C_14_H_10_Br_2_N_4_: C, 42.67; H, 2.56; N, 14.22; found: C, 42.91; H, 2.72; N, 14.36.

#### 3.2.27. 5-Bromo-3-((2-(6-chloropyridin-2-yl)hydrazono)methyl)-1*H*-indole (**27**)

The general procedure was used with 5-bromoindole-3-carboxaldehyde (0.25 g, 1.1 mmol) and 2-chloro-6-hydrazinopyridine (0.16 g, 1.1 mmol) to obtain pure compound **27** as a beige solid (0.26 g, 0.7 mmol, 68% yield); mp (EtOH) 214–216 °C; Rf (EtOAc:Cyclohexane 1:1) = 0.3. ^1^H NMR (DMSO-*d_6_*, 500 MHz) δ ppm 6.73 (d, *J* = 7.5 Hz, 1H, Ar*H*), 7.12 (d, *J* = 7.5 Hz, 1H, Ar*H*), 7.33 (dd, *J* = 8.5, 2.0 Hz, 1H, Ar*H*), 7.41 (d, *J* = 8.5 Hz, 1H, Ar*H*), 7.72 (d, *J* = 7.5 Hz, 1H, Ar*H*), 7.79 (d, *J* = 2.0 Hz, 1H, Ar*H*), 8.22 (s, 1H, =C*H*), 8.32 (d, *J* = 2.0 Hz, 1H, Ar*H*), 10.95 (s, 1H, N*H*), 11.67 (s, 1H, N*H*). ^13^C NMR (DMSO-*d_6_*, 125 MHz) δ ppm 103.7 (CH), 111.6 (C), 112.5 (CH), 112.8 (C), 113.9 (CH), 123.6 (CH), 125.0 (CH), 125.8 (C), 130.3 (CH), 135.8 (C), 138.2 (=CH), 141.1 (CH), 148.3 (C), 157.4 (C). IR *ν* (cm^−1^): 3342, 3160, 1622, 1593, 1556, 1531, 1474, 1425, 1317, 1283, 1234, 1097, 983, 920, 876, 773, 612. Elemental analysis calcd (%) for C_14_H_10_BrClN_4_: C, 48.10; H, 2.88; N, 16.03; found: C, 48.42; H, 2.96; N, 16.27.

### 3.3. Cell Proliferation Assay

Compounds **1**–**27** were tested on a panel of 60 human cancer cell lines at the National Cancer Institute, Germantown, MD [41]. The cytotoxicity studies were conducted using a 48h exposure protocol using the sulforhodamine B assay [42,43].

## Data Availability

The data presented in this study are available on request from the corresponding authors. Appendix A include copies of NMR spectra of all synthesized compounds.

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
