# Peer review of "7-Chloroquinolinehydrazones as First-in-Class Anticancer Experimental Drugs in the NCI-60 Screen among Different Investigated Series of Aryl, Quinoline, Pyridine, Benzothiazole and Imidazolehydrazones"

_pharmaceuticals, 2023, doi:10.3390/ph16050691_

Round 1
Reviewer 1 Report
In this work, 27 hydrazones were synthesized and screened for their potential to inhibit the NCI-60 cancer cell panel. The study resulted in a new collection of 7-chloroquinoline hydrazones with promising potential for further development of compounds for oncology. The research content belongs to the hot topic of drug research. Therefore, I think that this article is suitable for publication after minor revision.
Add a description of the research background in the abstract section.
Nowadays, the middle character "–" is preferred between the numbers. Please unify.
Table 2, Missing annotation c, excessive annotation f.
In additional materials, please calibrate all spectra with TMS (or residual solvent), provide calibration values in the general part and in each spectrum separately.
How to determine the configuration of compounds with cis trans isomerism.
The signal to noise ratio is poor in 13C-spectra in Figures S6b and S7b. Please re-run the spectra.
very good.
Author Response
On behalf of all authors of the manuscript, I would like to thank the reviewers for their comments which contributed to the improvement of the current version of the manuscript. We have addressed all the comments and suggestions indicated by the reviewers.
Reviewer 1:
“In this work, 27 hydrazones were synthesized and screened for their potential to inhibit the NCI-60 cancer cell panel. The study resulted in a new collection of 7-chloroquinoline hydrazones with promising potential for further development of compounds for oncology. The research content belongs to the hot topic of drug research. Therefore, I think that this article is suitable for publication after minor revision.”
“Add a description of the research background in the abstract section.”
The research background has been added in the abstract.
Nowadays, the middle character "–" is preferred between the numbers. Please unify.
This has been corrected throughout the manuscript.
Table 2, Missing annotation c, excessive annotation f.
The missing annotation c has been added and annotation f has deleted in Table 2. The annotation f is in fact annotation d.
In additional materials, please calibrate all spectra with TMS (or residual solvent), provide calibration values in the general part and in each spectrum separately.
All spectra have been realized using deuterated solvents (CDCl3 99.8%D +0.03% TMS V/V or DMSO-d6 99.8%D + 0.03% TMS V/V), purchased from Eurisotop, Saint-Aubin, France. The calibration has been realized automatically, using TMS pic as the 0.00 ppm value. This information has been added in the materials and methods section.
How to determine the configuration of compounds with cis trans isomerism.
All hydrazones were obtained as trans isomers. In a previous paper (Negru, G. et al, Molecules, 2021 26 5681), we have reported X ray structures for similar hydrazones. For example, we have reported the X-ray structure of the 2-chloro analogue of hydrazone 1 from the current manuscript, confirming the E-isomerism.
The signal to noise ratio is poor in 13C-spectra in Figures S6b and S7b. Please re-run the spectra.
Compounds bearing the 7-chloroquinoline unit generally displayed a poor signal to noise ratio in standard conditions. We have rerun spectra and, for example, for compound 7, we have run a 24-hour acquisition for the 13C NMR spectrum with 12000 scans instead of standard 1024 scans for other compounds. The signal to noise ratio was slightly improved but remained however poor. This confirmed that 7-chloroquinoline unit induced that. The new spectra have been added in the current version of the manuscript.
Reviewer 2 Report
New hydrazones with quinoline, pyridine, benzothiazole, and imidazole moieties have been synthesized and tested against 60 cancer cell lines for their cytotoxic potential. In the current investigation, 7-chloroquinolinehydrazones exhibited the highest cytotoxic activity with submicromolar GI50 values against a large panel of cell lines from nine tumor types. As anticancer agents, dihydroimidazole derivatives 3–5 were entirely ineffective and do not merit further research. Similarly, hydra-zones containing two differently substituted aryl units 1 and 2 were ineffective at inhibiting cancer cell proliferation. In the present investigation, the combination of two heterocyclic units proved to be the most tolerable and effective pharmacomodulation.
The result and discussion section of this study is extremely unclear, and there is no sub-section.
It is strongly advised to create sub sections.
There is no discussion of the synthesis section, which is vital and cannot be overlooked.
The authors have attempted to explain SAR in the discussion, but it is unclear; therefore, it is recommended to create a (Figure) general chemical structure of the backbone, with arrows pointing to each component and a concise summary of each section.
When discussing biological activities, there is no distinct discussion based on cell line types.
A distinct study should be conducted to demonstrate the viability of normal cells.
“Concerning the 1-methyl-5-methoxyindole unit (B-ring of target com-pounds): 1) the replacement of the indole unit with an aromatic unit (3-chloro-4-methoxyphenyl in hydrazone 6 and 4-hydroxy-3,5-dimethoxyphenyl in hy-drazone 7) conserved antitumor efficiency (GI50 values in the micromolar range) but resulted in diminished antitumor effect compared to indole-hydra-zones 16 and 23 (Table 2)” Provide the related reference of similar compounds or similar substitution pattern for different compounds from literature showing such behavior to give some comparison.
“In order to rule out the potential influence of the overall volume of the molecule 8 on the lack of biological effect, smaller size phenothiazine hydrazones have been synthesized (hydrazones 9-12, Scheme 1). None of these hydrazones were active in the NCI-60 screen” Provide the related reference of similar compounds or similar substitution pattern for different compounds from literature showing such behavior to give some comparison.
“The absence of the 5-methoxy substitu-ent in hydrazone 13 conserved the antitumor efficiency on a large number of cell lines but displayed slightly reduced efficiency compared with 5-methoxy-analogue hydrazone 16, especially on ovarian, renal, prostate and breast can-cer cell lines (Table 2)” Provide the related reference of similar compounds or similar substitution pattern for different compounds from literature showing such behavior to give some comparison.
There are many typo errors and also fluency issues.
Author Response
Reviewer 2:
The result and discussion section of this study is extremely unclear, and there is no sub-section.
It is strongly advised to create sub sections.
Subsections have been added in the “results and discussion” section.
There is no discussion of the synthesis section, which is vital and cannot be overlooked.
The reviewer is right. The synthesis section is short, but the essential information has been provided, including structures of both starting materials and products obtained. This chemistry is not complex, this kind of condensation is well known, and all compounds have been fully characterized in the manuscript and spectra have been provided in the supplementary material. No by-products were obtained. We have added the information concerning the E-isomerism for final hydrazones.
The authors have attempted to explain SAR in the discussion, but it is unclear; therefore, it is recommended to create a (Figure) general chemical structure of the backbone, with arrows pointing to each component and a concise summary of each section.
A global visualization of the structure-activity relationships for hydrazones obtained in this study has been proposed in current Figure 3 of the revised manuscript.
When discussing biological activities, there is no distinct discussion based on cell line types.
A distinct study should be conducted to demonstrate the viability of normal cells.
This is a perspective that we will consider in due course after identifying collaborators with this expertise. The National Cancer Institute is not providing this type of evaluation on normal cells (e.g. HEK-293, HK2, NRK-52E, etc).
“Concerning the 1-methyl-5-methoxyindole unit (B-ring of target com-pounds): 1) the replacement of the indole unit with an aromatic unit (3-chloro-4-methoxyphenyl in hydrazone 6 and 4-hydroxy-3,5-dimethoxyphenyl in hy-drazone 7) conserved antitumor efficiency (GI50 values in the micromolar range) but resulted in diminished antitumor effect compared to indole-hydra-zones 16 and 23 (Table 2)” Provide the related reference of similar compounds or similar substitution pattern for different compounds from literature showing such behavior to give some comparison.
This type of biological profile has been observed for tubulin polymerization inhibitors, but the bridge was not a hydrazone group. It was either a carbonyl, a sulfur, a sulfone, or ethylenic. Consequently, a direct comparison doesn’t seem appropriate. Moreover, the mechanism of action seems to be different than those inhibitors. The NCI-Compare analysis suggested a mechanism as DNA-intercalatants for hydrazones from this study and not tubulin polymerization inhibitors.
“In order to rule out the potential influence of the overall volume of the molecule 8 on the lack of biological effect, smaller size phenothiazine hydrazones have been synthesized (hydrazones 9-12, Scheme 1). None of these hydrazones were active in the NCI-60 screen” Provide the related reference of similar compounds or similar substitution pattern for different compounds from literature showing such behavior to give some comparison.
We have run a SciFinder research on phenothiazine-3-yl-containing hydrazones but we only found derivatives with properties as hole-transporting materials (Synthetic Metals (2022), 287, 117057, hemistrySelect (2017), 2(34), 11307-11313, etc).
“The absence of the 5-methoxy substitu-ent in hydrazone 13 conserved the antitumor efficiency on a large number of cell lines but displayed slightly reduced efficiency compared with 5-methoxy-analogue hydrazone 16, especially on ovarian, renal, prostate and breast can-cer cell lines (Table 2)” Provide the related reference of similar compounds or similar substitution pattern for different compounds from literature showing such behavior to give some comparison.
We only found molecule II (Figure 1) with similar substitution pattern in the literature (a pyrrole unit instead of the indole). The latter compound bearing an indole unit proved to be more active than the pyrrole congener. Similar compounds have been recently described by Purgatorio et al. in 2020 (Molecules 2020, 25(23), 5773) already cited in the manuscript as reference 40, but their activity was not evaluated for cytotoxicity.

Reviewer 3 Report
The research article entitled "7-Chloroquinolinehydrazones as first-in-class anticancer experimental drug in the NCI-60 screen among different investigated series of aryl, quinoline, pyridine, benzothi-azole and imidazolehydrazones" submitted by authors is an average scientific research problem. There is not much novelty in terms of new molecule design for antitumoral activity, instead authors made bunch of compounds (27) with different aryl and hetero aryl moieties around both sides of hydrozone core structure.
However, they established proper SAR and explained very well throughout the paper and provided significance of functional groups on both sides of aryl and hetero aryl groups. Provided detailed experimental data. This work has some significance in anti-cancer research and highly useful for future directions for the development of potent anticancer agents.
Minor problems:
1. Keywords are missing
2. Need to improve introduction part with more information
3. Check minor spelling mistakes - page-4 at the end of the page- please change "electro-withdrawing" to electron-withdrawing. such small spelling mistakes needs corrected.
4. At Materials and methods section - at line 60 - please specify the concentration of aq NaOH and at line 68 - please specify volume or concentration of ethanol.
After addressing above mentioned minor corrections, I recommend this article to publish in Pharmaceutical journal.
Quality of English is not clear. Need to correct small spelling and grammar mistakes throughout out the manuscript.
Author Response
Reviewer 3:
The research article entitled "7-Chloroquinolinehydrazones as first-in-class anticancer experimental drug in the NCI-60 screen among different investigated series of aryl, quinoline, pyridine, benzothi-azole and imidazolehydrazones" submitted by authors is an average scientific research problem. There is not much novelty in terms of new molecule design for antitumoral activity, instead authors made bunch of compounds (27) with different aryl and hetero aryl moieties around both sides of hydrozone core structure.
However, they established proper SAR and explained very well throughout the paper and provided significance of functional groups on both sides of aryl and hetero aryl groups. Provided detailed experimental data. This work has some significance in anti-cancer research and highly useful for future directions for the development of potent anticancer agents.
Minor problems:
- Keywords are missing
Keywords (hydrazone; 7-chloroquinoline; NCI panel; antitumor; pharmacomodulation; SAR) have been added in the manuscript.
- Need to improve introduction part with more information
We have improved the beginning of the manuscript by adding the research background in the abstract and the content of the manuscript by adding a global vision (Figure 3) of the generated structure-activity relationships on this hydrazone series. Even if it is concise, we believe that all essential information for the manuscript is described in the introduction and well documented (33 references). Given the fact that hydrazones with 7-chloroqinoline unit and other heterocycle on the part of the molecule are scarce in the literature, with all due respect for the reviewer’s suggestion, we chose to keep it concise.
- Check minor spelling mistakes - page-4 at the end of the page- please change "electro-withdrawing" to electron-withdrawing. such small spelling mistakes needs corrected.
These mistakes have been corrected.
- At Materials and methods section - at line 60 - please specify the concentration of aq NaOH and at line 68 - please specify volume or concentration of ethanol.
The concentration of aqueous NaOH was 32% and has been specified in the current version of the manuscript. The volume of ethanol was also specified in the revised manuscript.
After addressing above mentioned minor corrections, I recommend this article to publish in Pharmaceutical journal.

Round 2
Reviewer 2 Report
Authors have revised the manuscript as advised.
Final check required.